



# Improving volcanic ash predictions with the HYSPLIT dispersion model by assimilating MODIS satellite retrievals

Tianfeng Chai[1,2], Alice Crawford[1,2], Barbara Stunder[1], Michael J. Pavolonis[3], Roland Draxler[1,4], and Ariel Stein[1]

[1]NOAA Air Resources Laboratory (ARL), NOAA Center for Weather and Climate Prediction, 5830 University Research Court, College Park, MD 20740, USA
[2]Cooperative Institute for Climate and Satellites, University of Maryland, College Park, MD 20740, USA
[3]NOAA Center for Satellite Applications and Research, Madison, WI, USA
[4]Independent contractor

*Correspondence to:* Tianfeng Chai (Tianfeng.Chai@noaa.gov)

**Abstract.** Currently NOAA's National Weather Service (NWS) runs the HYSPLIT dispersion model with a unit mass release rate to predict the transport and dispersion of volcanic ash. The model predictions provide information for the Volcanic Ash Advisory Centers (VAAC) to issue advisories to meteorological watch offices, area control centers, flight information centers, and others. This research aims provide quantitative forecasts of ash distributions generated by objectively and optimally esti-
5  mating the volcanic ash source strengths, vertical distribution and temporal variations using an observation-modeling inversion technique. In this top-down approach, a cost functional is defined to mainly quantify the differences between model predictions and the satellite measurements of column integrated ash concentrations, weighted by the model and observation uncertainties. Minimizing this cost functional by adjusting the sources provides the volcanic ash emission estimates. As an example, MODIS (MOderate Resolution Imaging Spectroradiometer) satellite retrievals of the 2008 Kasatochi volcanic ash clouds are used to
10  test the HYSPLIT volcanic ash inverse system. Because the satellite retrievals include the ash cloud top height but not the bottom height, three options for matching the model concentrations to the observed mass loadings are tested. Although the emission estimates vary significantly with different options the subsequent model predictions with the different release estimates all show decent skill when evaluated against the unassimilated satellite observations at later times. Among the three options, integrating over three model layers yields slightly better results than integrating from the surface up to the volcanic ash
15  cloud top or using a single model layer. Inverse tests also show that including the ash-free region to constrain the model is not beneficial for the current case. In addition, extra constraints to the source terms can be given by explicitly enforcing "no-ash" for the atmosphere columns above or below the observed ash cloud top height. However, in this case such extra constraints are not helpful for the inverse modeling. It is also found that simultaneously assimilating observations at different times produces better hindcasts than only assimilating the most recent observations.





## 1 Introduction

Large amounts of ash particles are produced during violent volcanic eruptions. After the initial ejection momentum carrying them upwards, ash particles rise buoyantly into the atmosphere. Then volcanic ash travels away from the volcano following the atmospheric flow. Fine ash particles may remain in the atmosphere for days to weeks or longer and can travel thousands of miles away from the source. They have severe adverse impacts on the aviation industry, human and animal health, agriculture, buildings, and other infrastructure. To help prepare for and mitigate such impacts, it is important to not only monitor but also forecast the volcanic ash transport and dispersion.

Starting from a memorandum of understanding (MOU) signed between the United States National Oceanic and Atmospheric Administration (NOAA) and the Federal Aviation Administration (FAA) in December 1988, the NOAA Air Resources Laboratory (ARL) developed a Volcanic Ash Forecast Transport And Dispersion (VAFTAD) model for emergency response focusing on hazards to aircraft flight operations (Heffter and Stunder, 1993). Currently NOAA's National Weather Service (NWS) runs the HYSPLIT dispersion model (Draxler and Hess, 1997; Stein et al., 2015a) with a unit mass release rate to qualitatively predict the transport and dispersion of volcanic ash. The model predictions provide important information for the Volcanic Ash Advisory Centers (VAAC) to issue advisories to meteorological watch offices, area control centers, flight information centers, and others.

In order to quantitatively predict volcanic ash, realistic source parameters need to be assigned to the volcanic ash transport and dispersion models. Mastin et al. (2009) compiled a list of eruptions which had well-constrained source parameters. They found that the mass fraction of debris finer than $63\mu$m (m63) could vary by nearly two orders of magnitude between small basaltic eruptions ($\sim 0.01$) and large silicic ones ($> 0.5$). Default source parameters were assigned to the world's more than 1500 volcanoes. They may be used for ash-cloud modeling when few observations are available in the event of an eruption.

With the advancement of remote sensing techniques, satellites have played an important role in detecting and monitoring volcanic ash clouds (Seftor et al., 1997; Ellrod et al., 2003; Pergola et al., 2004). An automated volcanic ash cloud detection system has been developed and continuously improved (Pavolonis et al., 2006, 2013, 2015a, b). In addition to detecting and monitoring ash cloud, satellite measurements allow many ash cloud characteristics to be quantified. For instance, Wen and Rose (1994) used two-band data from NOAA Advanced Very high Resolution Radiometer (AVHRR) to retrieve total mass of a volcanic ash cloud from the August 19, 1992 Crater Peak/Spurr Volcano, Alaska eruption. Using multi-spectral satellite data from the AVHRR-2 and ATSR-2 instruments, Prata and Grant (2001) provided a quantitative analysis of several properties of the Mt Ruapehu, New Zealand, ash cloud, including mass loading, cloud height, ash cloud thickness, and particle radius. The quantified ash cloud parameters can be directly inserted into transport and dispersion models as 'virtual sources' far from the vent. Wilkins et al. (2014, 2016) applied this technique to the eruption of Eyjafjallajökull in 2010 using infrared (IR) satellite imagery and the NAME model. It was also applied by Crawford et al. (2016) to the 2008 Kasatochi eruption using the HYSPLIT model.

Under a general data assimilation and inverse modeling framework, satellite measurements can be used to constrain the model and estimate emission parameters using various techniques. For instance, Stohl et al. (2011) applied an inversion scheme





to the Eyjafjallajökull eruption using a Lagrangian dispersion model with satellite data and demonstrated the effectiveness of the method to yield better quantitative volcanic ash predictions. Schmehl et al. (2012) proposed a variational technique that uses a genetic algorithm (GA) to assimilate satellite data to determine emission rates and the steering winds. A HYSPLIT inverse system based on a four-dimensional variational data assimilation approach has been built and successfully applied to estimate

the cesium-137 releases from the Fukushima Daiichi Nuclear Power plant accident in 2011 (Chai et al., 2015). The present work further develops on the HYSPLIT inversion system to estimate the time- and height-resolved volcanic ash emission rate by assimilating satellite observations of volcanic ash. The system is tested with the 2008 Kasatochi eruption using the satellite retrievals from passive IR sensors.

The paper is organized as follows. Section 2 describes the satellite observations of volcanic ash, HYSPLIT model and

configuration, and the inverse modeling methodology. Section 3 presents emission inversion results and Section 4 discusses the corresponding volcanic ash forecasts with the estimated source terms. A summary is given in Section 5.

## 2 Methodology

### 2.1 Satellite observations

The volcanic ash observations are based on MODIS retrievals from Terra and Aqua satellites. They include ash mass loading,

cloud top height, and effective particle radius. Pavolonis et al. (2013, 2015a, b) described the details of the retrieval methodology and how the ash cloud observations are derived from the retrieved parameters such as radiative temperature and emissivity. Here volcanic ash observations of the 2008 Kasatochi eruption at five different instances are utilized. The observations were projected to a latitude-longitude grid with a resolution of $0.05^o$ in latitude and in $0.1^o$ longitude. Figure 1 shows volcanic ash mass loadings and ash cloud top heights of five granules. Each granule contains 6 minutes of data and it covers an area

of approximately 1500 km along the orbit and 1650 km wide. Note that the satellite observations outside the shown domain are discarded. As the discarded data are mostly located upwind of the volcano vent, they are not expected to provide useful information to estimate the source strength. The places where satellite retrievals did not detect existence of ash show zero mass loading. It will be shown later that such ash free regions may be used along with the observed ash cloud to constrain the dispersion model. Table 1 shows the observation time and number of grid cells with and without ash detected for each granule.

It is seen that the clear regions dominate the satellite observations. Integrated mass loadings based on the satellite data are also listed in Table 1. They decrease from $9.68\times10^8$ kg for the first granule (G1) to $3.25\times10^8$ kg for the last granule (G5). This probably reflects the gradual loss of the total volcanic ash mass due to deposition. Note the total mass is likely slightly underestimated for the second granule (G2) where the satellite lost sight of the eastern edge of the ash cloud.

### 2.2 HYSPLIT model configuration

In this study, volcanic ash transport and dispersion are modeled using the HYSPLIT model (Draxler and Hess, 1997, 1998; Stein et al., 2015a). A large number of three-dimensional Lagrangian particles are released from the source location and pas-



**Figure 1.** MODIS volcanic ash mass loadings (left) and ash cloud top height(right) listed from top to bottom following their observation time (see Table 1 for detail). "+" shows the location of Kasatochi volcano (52.1714$^o$N, 175.5183$^o$W). Note that the satellite observations to the left of the map domain are not used in this paper.





**Table 1.** Description of MODIS ash cloud observations. "Ash cells" and "clear cells" show number of grid cells with and without ash detected, respectively. Total mass is obtained by integrating mass loadings over the observed region.

|    | Observation time | Ash cells | Clear cells | Total mass (kg) |
|----|------------------|-----------|-------------|-----------------|
| G1 | 1340Z on 8 August, 2008 | 3778 | 92230 | $9.68 \times 10^8$ |
| G2 | 0050Z on 9 August, 2008 | 9604 | 56161 | $6.69 \times 10^8$ |
| G3 | 1250Z on 9 August, 2008 | 13226 | 107104 | $5.37 \times 10^8$ |
| G4 | 0000Z on 10 August, 2008 | 13876 | 98686 | $3.72 \times 10^8$ |
| G5 | 1150Z on 10 August, 2008 | 15088 | 100211 | $3.25 \times 10^8$ |

sively follow the wind afterward. A random component based on local stability is added to the mean advection velocity in each of the three-dimensional wind component directions to simulate the dispersion. Ash concentrations are computed by summing each particle's mass as it passes over a concentration grid cell and dividing the result by the cell's volume.

Both NOAA's Global Data Assimilation System (GDAS) (Kleist et al., 2009) and the European Centre for Medium-Range Weather Forecasts (ECMWF) ERA Interim global atmospheric reanalysis (Dee et al., 2011) were used as inputs for HYSPLIT. The basic information of the two data sets is listed in Table 2. The concentration grid is set at $0.05^o$ resolution in latitude and $0.1^o$ in longitude with a vertical spacing of 2 km extending from the surface to 20 km.

A total of 290 independent HYSPLIT simulations were run with a unit emission rate released from all possible combinations of 29 different hours from 19Z, August 7, 2008 to 23Z, August 8, 2008, and 10 different 2000m layers. Note that at the first layer, particles are released from the top of the vent, 300 m above sea level to 2000m, while at other layers particle releases are uniformly distributed throughout the layer at the center of the grid as a line source. In each simulation, particles of four different sizes are released as different pollutants. At all release time and height combinations, the contributions to the total mass are assumed constant, at 0.8%, 6.8%, 25.4%, and 67.0% for particle sizes of 0.6 $\mu$m, 2.0 $\mu$m, 6.0 $\mu$m, 20.0 $\mu$m, respectively.

**Table 2.** Description of GDAS and ECMWF meteorological data.

| Data set | Horizontal resolution | Vertical pressure levels | Output interval |
|----------|----------------------|--------------------------|-----------------|
| GDAS | $1^o \times 1^o$ | every 25 hPa from 1000 to 900 hPa, every 50 hPa from 900 to 50 hPa, and 20 hPa | 3 hours |
| ECMWF | $0.75^o \times 0.75^o$ | every 25 hPa from 1000 to 750 hPa, every 50 hPa from 750 to 250 hPa, every 25 hPa from 250 to 100 hPa, 70 hPa, 50 hPa, 30 hPa, and 20 hPa | 6 hours |





## 2.3 Matching model to observations

As shown in Section 2.1, satellite observations provide ash mass loadings and ash cloud top heights after detecting ash. There are several options to construct the model counterparts for observed ash cloud mass loadings. Three different options are tested here. In the first option, model volcanic ash concentrations from the ground or sea level up to the model layer where the cloud

top height resides are integrated to calculate the mass loadings by the model simulation. In the second option, the single model layer where the retrieved cloud top height resides is used to construct the mass loadings. Integrating over three layers, i.e. from one layer below to one layer above the cloud top layer, is the third option to be tested.

When ash is not detected, grid cells are flagged as clear or ash-free. This is equivalent to zero mass loading and infinite cloud top height. The model counterpart is obtained by integrating simulated concentrations from the surface to the domain

top. Constraining the model simulation with these zero-value observations is expected to help remove spurious sources from which the transport and dispersion will likely generate additional ash clouds which are not observed.

At locations where ash is detected, the observations can be further exploited to provide additional constraints. As ash cloud top heights are provided along with the mass loadings, they indicate that no ash is above the cloud top. However, no information can be inferred for the region below the cloud top. As a result, each ash cell actually generates two pieces of information.

Besides the observed volcanic ash cloud mass loadings mentioned earlier, clear atmospheric columns above the cloud top is the other implicit piece of information that can be used in emission inversion as well. Similar to using zero-value observations at ash-free locations, the integrated mass loadings above the ash cloud top may also be used to filter out unlikely sources. When the "observed" ash cloud is assumed to be limited to one single model layer or three layers, it is also possible to add no-ash-below-cloud constraints in the inverse modeling. Although such constraints are based on an assumption that is not always true,

it will be tested nonetheless.

In addition to detected ash and clear cells, another scenario exists when satellite observations cannot provide positive or negative answers for ash detection, e.g., due to meteorological cloud obstruction. In such a case, no useful information can be used to constrain the model. For the 2008 Kasatochi eruption, overlying meteorological clouds were nearly absent and valid observations appear across the satellite swaths.

## 25  2.4 Transfer Coefficient Matrix (TCM)

A transfer coefficient matrix (TCM) of 290 columns can be generated using all or a subset of the re-gridded MODIS observations listed in Table 1 and the results of the 290 HYSPLIT simulations with unit emission. A transfer coefficient in the TCM is essentially the mass loadings at an observation point that the row represents resulted from a dispersion run with a unit emission that the column indicates.

Figure 2 shows the two-dimensional transfer coefficient matrices averaged over all ash pixels for five granules. As a transfer coefficient corresponds to the mass loadings resulted from a unit ash release rate, integrating over more model layers would produce larger transfer coefficients. It is clearly seen that the single layer option, shown as the middle column in Figure 2, has the averaged TCMs with the lowest values. Figure 2 also shows that integrating from surface up to ash cloud top layer





generally results in TCMs with the largest values among the three options. As the option to add over three layers (right column in Figure 2) includes a layer above the cloud top layer that is not included in option 1, transfer coefficients at the upper layers may have larger values. Note that a block of zero transfer coefficients after 10Z August 8 appear for G1. Ash releases after the observation time of G1 do not affect G1 observations. In addition, releases need time to travel to the observed location.

Figure 2 shows that, as expected, the averaged transfer coefficients tend to be smaller for later observations due to dispersion. The averaged TCMs using ECMWF meteorological data (not shown) are similar to the GDAS results shown here.

## 2.5   Emission Inversion

Following a general top-down approach, the unknown emission terms are obtained by searching for the emissions that would provide the model predictions which most closely match the observations. In the current application with the known volcano

location, the emission rates vary with time and release heights. With the potential emission time period divided into 29 hourly intervals and the release heights separated into 10 vertical layers, the discretized two-dimensional unknown emission has 290 components to be determined.

Similar to Chai et al. (2015), the unknown releases can be solved by minimizing a cost functional that integrates the differences between model predictions and observations, deviations of the final solution from the first guess (*a priori*), as well as

other relevant information written into penalty terms (Daley, 1991). For the current application, the cost functional $\mathcal{F}$ is defined as,

$$\mathcal{F} = \frac{1}{2} \sum_{i=1}^{M} \sum_{j=1}^{N} \frac{(q_{ij} - q_{ij}^b)^2}{\sigma_{ij}^2} + \frac{1}{2} \sum_{m=1}^{M} \frac{(a_m^h - a_m^o)^2}{\epsilon_m^2} \tag{1}$$

where $q_{ij}$ is the discretized two-dimensional emission over M=29 hours and N=10 layers. $q_{ij}^b$ is the first guess or *a priori* estimate and $\sigma_{ij}^2$ is the corresponding error variance. Note that we assume the uncertainties of the release at each time-height are independent of each other so that only the diagonal term $\sigma_{ij}^2$ of the typical *a priori* error covariance matrix appears in

Equation 1. We choose a small constant emission rate of $10^4$ g/hr ($\approx 2.8 \times 10^{-3} kg/s$) at all hours and layers as the first guess. Large uncertainties are given in the following tests to reflect the fact that little was known for the mass emission rates. $a_m^h$ and $a_m^o$ are the mass loadings simulated by HYSPLIT and retrieved by MODIS, respectively. The observations here refer to both the volcanic ash mass loadings for the ash cloud and the zero values for the ash-free regions. Zero mass loadings also include those calculated over the atmospheric columns above or below ash clouds as discussed earlier in Section 2.4. $\epsilon_m^2$ includes the

variances of the observational and representative errors. For simplicity, $\epsilon_m^2$ are referred as observational errors hereafter and are assumed to be uncorrelated with $\epsilon_m = 0.50 \times a_m^o + 0.3 \ g/m^2$. No smoothness penalty term is included in the cost functional because of the abrupt nature of the volcanic eruptions. A large-scale bound-constrained limited-memory quasi-Newton code, L-BFGS-B (Zhu et al., 1997) is used to minimize the cost functional $\mathcal{F}$ defined in Equation 1. The maximum number of cost functional evaluations is set as 250 for cases in Section 3 and 2500 for those in Section 4. To ensure non-negative $q_{ij}$

solutions from the optimization, $q_{ij}$ is converted to $ln(q_{ij})$ as input to the L-BFGS-B routine. An alternative to this is enforcing





**Figure 2.** Averaged TCMs with three different options in calculating model mass loadings (Column 1: integrating from surface to cloud top; Column 2: calculated for a single layer where the cloud top height resides; Column 3: integrating over three layers centered at the cloud top layer). Rows 1-5 (from top to bottom) correspond to observations G1-5.





the $q_{ij} \geq 0$ with lower bounds enabled by the L-BFGS-B routine. Chai et al. (2015) provides a detailed discussion on the conversion of control and metric variables. Although they showed that using logarithmic concentration differences in the cost functional performed better than directly using concentration differences in their application, the logarithmic conversion on the metric variable $a_m$ is not beneficial for the current application. It is because the range of the volcanic ash mass loadings here

is much smaller than that of the Cs-137 air concentrations encountered in their application. In addition, the utilization of zero mass loadings in many ash-free regions prohibits using $ln(a_m^o)$. In this study, the mass loadings are directly compared in the cost functional without logarithmic conversion.

## 3  Emission estimates

The emission estimates obtained by minimizing the the cost functional $\mathcal{F}$ introduced in Equation 1 highly depend on the

uncertainties given to the *a priori* and observations. Sensitivity tests are first performed by changing the magnitudes of the *a priori* error variances while the observational error estimation is fixed. Chai et al. (2015) demonstrated that the emission inversion results were not sensitive to the first guess of the emissions when large uncertainties are presumed.

In the sensitivity tests, ash cloud data at G1 and G2 are assimilated. Large *a priori* error variances are presumed, with $\sigma_{ij} \approx 10^{12}$ g/hr ($\approx 2.8 \times 10^5$ kg/s) and $\sigma_{ij} \approx 10^{16}$ g/hr ($\approx 2.8 \times 10^9$ kg/s). Figure 3 shows that the emission inversion results

are slightly different from each other when the *a priori* errors are assumed differently, as expected. However, similar patterns are apparent for both cases with the different *a priori* error variances. A peak release greater than 5000 kg/s is observed at 04Z August 8, 2008 at the 6–8 km layer for both cases. This demonstrates that the emission estimates are most decided by the satellite data when *a priori* errors are assumed large enough. For the following tests, the *a priori* error variances are set as $\sigma_{ij} \approx 10^{12}$ g/hr ($\approx 2.8 \times 10^5$ kg/s).

Waythomas et al. (2010) characterize the eruption by three major explosive events and two smaller events. Events 1 and 2 started at 2201Z on Aug. 7 and 0150Z on Aug. 8, respectively. These two events reached 14 km and produced water-rich but ash-poor clouds. Event 3 happened at 0435Z on Aug. 8. It generated ash-rich cloud that rose up to 18 km. About 16 hours of continuous ash emission was punctuated with events 4 and 5 at 0712Z and 1142Z on Aug. 8.

In the above cases, the HYSPLIT simulated mass loadings were calculated by integrating from the surface to ash cloud top

heights at the ash cells. Figure 4 shows the emission estimates using all three options in calculating model mass loadings. The emission results are significantly different with different options. For the case where the model counterparts of the satellite mass loadings are obtained by integrating from surface to cloud top, the ash releases started at 01Z, August 8, 2008 from the 8–10 km layer. The emissions lasted for four hours and extended to multiple layers, reaching up to the 14–16 km layer, and down to the 4–6 km layer. After 1 hour without ash, moderate volcanic ash releases continued for six hours until 12Z on

August 8, and mainly between 8-16 km. A small ash emission of less than 80 kg/s is seen at the 12–14 km layer starting at 15Z for 1 hour. If the model mass loadings are obtained by only considering a single layer where the cloud top height resides, the resulting release source terms are limited to layers between 12–16 km. The ash releases started at 03Z, August 8, and lasted for three hours before resuming again two hours later. With emission on and off for the next two hours at the 14–16 km layer, the





**Figure 3.** Volcanic ash release results with different *a priori* error estimations (Top: $\sigma_{ij} \approx 2.8 \times 10^5$ kg/s; Bottom: $\sigma_{ij} \approx 2.8 \times 10^9$ kg/s). The TCMs for the emission inverse were generated using HYSPLIT runs with GDAS meteorological data. Only ash cells of the satellite data at G1 and G2 are used in the emission inverse. Model counterparts are obtained by integrating from surface to ash cloud top heights at ash cells.



ash release continued for 6 hours and peak at 14–15Z, August 8 at the 12–14 km layer. There is also an isolated emission point at the 14–16 km layer starting at 23Z, August 8 for an hour. In the last case where the model mass loadings are calculated by integrating over three layers centered at the cloud top layer, the ash releases are drastically different from the first two cases. The ash releases start much earlier, at 20Z, August 7 and the release heights are within the 14–18 km range. The release then

extended to more layers, but the main sources went lower. This lasted for 13 hours before stopping at 9Z on August 8. A second spurt of ash release started at 11Z from the 14–16 km layer and remained above 12 km before pausing again five hours later. Several weaker ash releases are found between 14–18 km layers at later times from 19Z on August 8 to 0Z on August 9.

The three emission estimates in Figure 4 do not reproduce the eruption as described by Waythomas et al. (2010), but manage to capture some characteristics of the eruption. Without information on the vertical profiles of the ash cloud, how the mass

loadings are interpreted greatly affect the release estimates, as shown by the drastic differences between the estimates shown in Figure 4. Thus, it is difficult to generate reliable and accurate actual volcanic ash emission estimates if the ash cloud vertical structures are undetermined. However, it will be shown later that such emission estimates can still help improve ash cloud forecasts.

## 4 Ash predictions with top-down emission estimates

A series of tests were performed to find the best inverse modeling setup which improved volcanic ash cloud forecasts the most. In Section 4.1, the evaluation metrics are described. In Section 4.2, the choices of calculating the model counterparts of the satellite mass loadings are compared. In Section 4.3, whether to use ash-free region to constrain the model is investigated. In Section 4.4, the effect of keeping older observations when newer observations become available is discussed.

### 4.1 Evaluation metrics

For model evaluation, total column mass loadings are constructed by integrating predicted concentrations from the surface to the domain top. They are used to compare with the satellite observations in each granule shown in Figure 1, including both ash and clear points. Using total column mass loadings instead of any of the options described in Section 2.4 aims to provide a fair comparison among the three options by avoiding the complexities associated with the vertical structures of the volcanic ash cloud. Note that Crawford et al. (2016) excluded mass below 2 km when integrating the model results to obtain the mass

loadings because the satellite retrieval is less sensitive to low-level ash. Such exclusion may improve the evaluation statistics but it is not expected to affect the inter-comparison between different model runs. Mean bias (MB), fractional bias (FB), root mean square error (RMSE), normalized RMSE (NRMSE), and Pearson correlation coefficient (R) are calculated. FB and NRMSE are scaled by the average of model and observation means. In addition, critical success index (CSI) defined below is calculated for ash detection.


$$CSI = \frac{N_{Hit}}{N_{FalseAlarm} + N_{Hit} + N_{Miss}} \qquad (2)$$





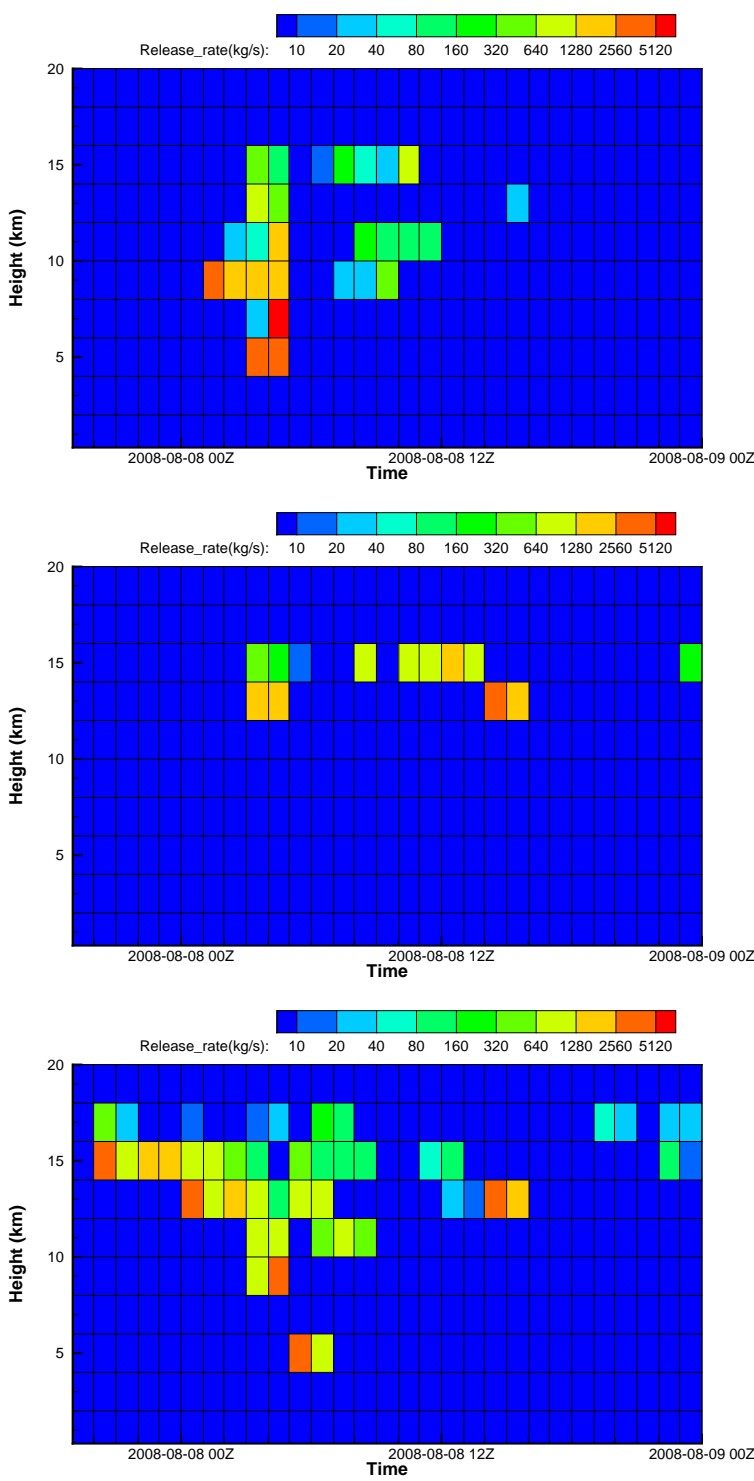

**Figure 4.** Volcanic ash release estimates with different options in model mass loading calculation. From top to bottom: integrating from surface to cloud top (same as Figure 3 top), calculated for a single layer where the cloud top height resides, and integrating over three layers centered at the cloud top layer.



A threshold of 0.1 $g/m^2$, the approximate lower limit of the MODIS satellite data set, is used to categorize ash existence for both model predictions and observations. $N_{Hit}$, $N_{FalseAlarm}$, and $N_{Miss}$ denote the numbers of grid points where ash is predicted and observed, ash is predicted but not observed, and ash is observed but not predicted by model, respectively.

Following Draxler (2006), Kolmogorov-Smirnov parameter (KSP) and "Rank" are calculated. KSP measures the largest difference between the cumulative distribution functions of the model predictions and observations. As shown in Equation 3, the "Rank" adds up four components which all range from 0 to 1. The larger "Rank" values indicate better overall performance of the model results.

$$Rank = R^2 + (1 - \frac{|FB|}{2}) + CSI + (1 - KSP) \tag{3}$$

## 4.2 Model mass loadings

The HYSPLIT predictions using the estimated source terms after assimilating G1 and G2 observations are evaluated against the satellite observations of G2, G3, G4, and G5, respectively. The three options to calculate the model ash mass loadings discussed earlier are employed in the inverse modeling. The statistics are listed in Table 3.

Comparing against the G2 observation, Table 3 shows that integrating over three model layers yields (option M1) slightly better results based on most statistics. It is true for cases with both GDAS and ECMWF meteorological fields. The advantage of M1 option is not apparent when comparing against other observations. Based on Rank, the ECMWF cases are better than the GDAS cases against G2, but the Ranks for ECMWF cases deteriorates faster with time, and become worse than the GDAS cases when model output is compared to G4 and G5 observations. The model predictions have the best statistics compared against G4 than against the other satellite granules (G2, G3, and G5). The case with GDAS meteorological fields and the three-layer mass loading option M1 has the best Rank of 3.02 (FB=0.04, R=0.72, CSI=0.62, KSP=0.10). If only G2 observations were assimilated, the model performance would be expected to peak when compared against G2. However, as both G1 and G2 observations are assimilated, this is no longer true. The effect of assimilating different observations will be discussed later in Section 4.4. Table 3 shows that the model tends to underestimate the ash mass loadings of G2 and G3 and then mostly overestimate the ash mass loadings of G4 and G5. It results in the best FB against G4 for GDAS cases and the best FB against G3 for ECMWF cases as the FB signs change. Since the volcanic ash will disperse with time, the average mass loadings get smaller. This is reflected in a basic trend of decreasing RMSEs with time although the NRMSEs slightly increase.

While different evaluation metrics may not always agree with each other, the overall performance parameter Rank provides a simplified way to compare model results. Only Ranks are listed and used to compare model predictions hereafter. Using HYSPLIT ensembles, Stein et al. (2015b) estimated the uncertainties for the Rank to be approximately 0.1.

## 4.3 Extra constraints

As discussed in Section 2.3, ash-free regions indicate zero mass loadings and infinite cloud top heights. Cloud top heights can also be used to enforce ash-free atmospheric columns above volcanic ash cloud. In addition, ash-free atmospheric columns




below the ash cloud may be assumed if an ash cloud thickness is estimated. Whether such extra constraints benefit the inverse modeling is tested here using the 22 inverse cases listed in Table 4. The Ranks evaluated against G2–5 are listed. It is found that when the additional constraints of including the clear pixels outside the ash cloud are used, the Ranks decrease. This holds true against G2-4, for all three mass loading calculation options, and for both sets of meteorological data. Two exceptions are found

against G5 for the ECMWF cases with the M0 and M1 options, in which Ranks increase from 2.17 to 2.32 and 2.28 to 2.38, respectively. Adding the extra constraints of a clear column above the ash cloud again generally causes a decrease in Rank. An exception is the ECMWF case with the M1 option (three model layers used for mass loading calculation) in which the extra "top" constraint results in a marginally better predictions evaluated against G5 (Rank 2.39 versus 2.38). When the constraints of clear column below ash cloud are further added for the M0 and M1 options, the ranks decrease significantly, especially for

the M0 option in which a single model layer is used to construct the model mass loadings.

## 4.4 Older observations

As newer observations become available, whether to include the older observation in the assimilation remains a question. Table 5 lists statistics of 10 cases evaluated against granules 2–5 using both GDAS and ECMWF fields. In the inverse modeling, only ash pixels were used and the model mass loadings are calculated by integrating over three layers centered at the cloud

top layer (M1 option). It is found that assimilating G2 and G1 yields greater Ranks when comparing against G3 and G4 observations than assimilating G2 alone. At G5, there is little difference between the two strategies. Note that assimilating G2 alone helps to get better statistics against the same observations than assimilating G1 and G2 at the same time, although this does not help the forecasts of G3 about 12 hours later.

After G3 is available, three strategies to utilize the available observations G1, G2 and G3 are tested. The results show that

assimilating G2 along with G3 observations achieve better forecasts at G4 and G5 moments than assimilating only G3. It is also found that including G1 in the assimilation does not make much difference. Again, the assimilation of G3 alone results in a closer match between model predictions and G3 observations, but the forecasts at later times are worse than if the earlier observations are also assimilated.

Figure 5 shows the comparison between MODIS observations and HYSPLIT simulations using the estimated source terms

obtained by assimilating G1, G2 and G3 with both GDAS and ECMWF meteorological fields, listed as the last two cases in Table 5. The simulated ash cloud corresponding to G1 are narrower than the satellite observations and the mass loading values are underestimated. Crawford et al. (2016) found that cylindrical source terms performed better than the line sources assumed here. Inverse modeling with cylindrical sources will be investigated in the future. The HYSPLIT simulations with both meteorological fields agree well with granules G2 and G3 and it is reflected by the high Rank vales (Table 5). This is

expected as the same observations were assimilated to obtain the ash release rates. Against G4, the model results capture the ash cloud locations and magnitudes very well for both cases. The case with GDAS inputs appears to have similar mass loading values as the observations while ECMWF case has a narrow ring inside the main ash cloud with higher values than the MODIS observations. In addition, the ECMWF case shows two tails while the GDAS case has only one tail resembling the MODIS observations. Both cases show tapering shapes of the tails which appear different from the satellite view. Against the later



observations of G5, HYSPLIT simulations start to deviate from the MODIS, as indicated by the lower Rank. Both GDAS and ECMWF simulations capture the ash cloud at the similar locations as observed by the satellite, but show smoother structures. It is speculated that meteorological fields with higher spatial and temporal resolutions might be able to improve the ash cloud predictions.

There were several lidar observations of the Kasatochi ash cloud provided by CALIPSO satellite (Winker et al., 2010; Kristiansen et al., 2010; Crawford et al., 2016). The HYSPLIT simulations shown in Figure 5 are also compared against the 532 nm backscatter vertical profiles along the three CALIPSO overpasses coincident with G1, G4, and G5. The comparisons reveal that both GDAS and ECMWF simulations captured the main ash cloud features at approximately the same location and altitude.

**5   Summary**

An inverse system based on HYSPLIT has been developed to solve the effective volcanic ash release rates as a function of time and height by assimilating satellite mass loadings and ash cloud top heights. The Kasatochi eruption in 2008 was used as an example to test and evaluate the current top-down system with both GDAS and ECMWF meteorological fields.

    When quantifying the differences between the model predictions and the satellite observations, the model counterparts can

be calculated differently using the 3-D model concentration results because the observed ash cloud bases are unknown. Three options to construct the model mass loadings, integrating volcanic ash concentrations from the surface up to the cloud top height or just using one or three model layers, are tested for this inverse system. It is found that the emission estimates vary significantly with different options. However, all the predictions with the different estimated release rates show decent skill when evaluated against the unassimilated satellite observations at later times. The option of integrating over three model layers

yields slightly better results than integrating from surface up to the cloud top or using a single model layer.

    The extra constraints of enforcing zero mass loading in the ash-free regions are tested with the inverse system. The model predictions using the emission estimates generated with such extra constraints are worse than those using the emission estimates generated by only assimilating the ash pixels. Additional "no-ash" constraints for the atmosphere columns above or below the observed ash cloud top height are found to further deteriorate the subsequent model predictions using the top-down emission

estimates.

    Assimilating multiple granules at different times prove to be beneficial. As new observations become available, the effect of one-day-old observations becomes marginal, but assimilating mass loadings from the most recent and those at about 12-hour earlier yield better results than only assimilating the most recent observations.

    The spatial and temporal resolutions of the meteorological fields may need improvement for future studies. The line source

assumed here can be replaced by more realistic cylindrical sources in the future. A simple particle size distribution with four different particle sizes is used at all release height and time. With MODIS effective radius available, a more realistic way to represent the particle size distribution can be explored.





**Figure 5.** Volcanic ash mass loadings from MODIS (left) and HYSPLIT simulations with GDAS (center) and ECMWF (right). From top to bottom following their observation time (see Table 1 for detail). "+" shows the location of Kasatochi volcano ($52.1714^o$N, $175.5183^o$W). White areas indicate regions outside satellite granules for MODIS observations. For HYSPLIT simulations, the white areas indicate zero mass loadings in order to reveal the ash cloud boundaries. The ash release rates for the HYSPLIT simulations were obtained by assimilating granules G1,G2,and G3. In the inverse modeling, only ash pixels were used and the model mass loadings are calculated by integrating over three layers centered at the cloud top layer.





*Author contributions.*

Tianfeng Chai wrote the inverse system code and conducted the inverse tests. Alice Crawford performed the HYSPLIT simulations with unit release rates. Barbara Stunder provided significant inputs on the volcanic ash modeling. Michael J. Pavolonis produced the satellite retrivals. Roland Draxler and Ariel Stein provided guidance on the HYSPLIT modelling. All authors contributed in writing the manuscruipt 5 and Tianfeng Chai wrote most of it.

*Acknowledgements.* This study was supported by NOAA grant NA09NES4400006 (Cooperative Institute for Climate and Satellites-CICS) at the NOAA Air Resources Laboratory in collaboration with the University of Maryland.





**Table 3.** Evaluation statistics against G2, G3, G4, and G5 observations for cases with different ways to calculate model mass loadings. G1 and G2 are assimilated for all cases listed here. MET: meteorological inputs. OBS: satellite observations used for evaluation. ML(Mass loading): MA, integrating from surface to cloud top; M0, calculated for a single layer where the cloud top height resides; M1, integrating over three layers centered at the cloud top layer. MB: mean bias; FB: fractional bias; RMSE: root mean square error; NRMSE: normalized RMSE; R: Pearson correlation coefficient; CSI: critical success index; KSP: Kolmogorov-Smirnov parameter. Rank is defined in Equation 3.

| MET | OBS | ML | MB ($g/m^2$) | FB | RMSE ($g/m^2$) | NRMSE | R | CSI | KSP | Rank |
|---|---|---|---|---|---|---|---|---|---|---|
| | | MA | -0.09 | -0.45 | 0.63 | 2.98 | 0.60 | 0.52 | 0.05 | 2.61 |
| | G2 | M0 | -0.10 | -0.45 | 0.68 | 3.25 | 0.54 | 0.54 | 0.04 | 2.58 |
| | | M1 | -0.10 | -0.47 | 0.63 | 3.03 | 0.60 | 0.58 | 0.04 | 2.66 |
| | | MA | -0.04 | -0.38 | 0.28 | 3.07 | 0.64 | 0.55 | 0.05 | 2.72 |
| G | G3 | M0 | -0.03 | -0.28 | 0.33 | 3.40 | 0.60 | 0.59 | 0.04 | 2.77 |
| D | | M1 | -0.03 | -0.32 | 0.30 | 3.13 | 0.61 | 0.61 | 0.05 | 2.77 |
| A | | MA | -0.01 | -0.10 | 0.18 | 2.40 | 0.72 | 0.62 | 0.12 | 2.96 |
| S | G4 | M0 | 0.01 | 0.10 | 0.25 | 3.02 | 0.65 | 0.64 | 0.07 | 2.96 |
| | | M1 | 0.00 | 0.04 | 0.19 | 2.39 | 0.72 | 0.62 | 0.10 | 3.02 |
| | | MA | -0.01 | -0.09 | 0.21 | 3.21 | 0.43 | 0.43 | 0.23 | 2.34 |
| | G5 | M0 | 0.01 | 0.19 | 0.25 | 3.33 | 0.41 | 0.45 | 0.22 | 2.31 |
| | | M1 | 0.01 | 0.12 | 0.22 | 3.12 | 0.43 | 0.45 | 0.25 | 2.32 |
| | | MA | -0.06 | -0.26 | 0.61 | 2.67 | 0.66 | 0.53 | 0.03 | 2.81 |
| | G2 | M0 | -0.04 | -0.16 | 0.72 | 3.00 | 0.65 | 0.58 | 0.05 | 2.87 |
| | | M1 | -0.07 | -0.32 | 0.60 | 2.69 | 0.69 | 0.63 | 0.04 | 2.90 |
| E | | MA | -0.01 | -0.13 | 0.34 | 3.25 | 0.62 | 0.52 | 0.04 | 2.80 |
| C | G3 | M0 | 0.01 | 0.05 | 0.45 | 4.01 | 0.60 | 0.56 | 0.04 | 2.85 |
| M | | M1 | -0.02 | -0.15 | 0.35 | 3.40 | 0.61 | 0.55 | 0.04 | 2.80 |
| W | | MA | 0.01 | 0.16 | 0.28 | 3.21 | 0.68 | 0.55 | 0.13 | 2.80 |
| F | G4 | M0 | -0.07 | -0.32 | 0.60 | 2.69 | 0.69 | 0.63 | 0.04 | 2.90 |
| | | M1 | 0.02 | 0.18 | 0.34 | 3.87 | 0.63 | 0.56 | 0.08 | 2.78 |
| | | MA | 0.01 | 0.18 | 0.26 | 3.55 | 0.42 | 0.45 | 0.21 | 2.33 |
| | G5 | M0 | 0.05 | 0.51 | 0.37 | 4.17 | 0.43 | 0.44 | 0.20 | 2.17 |
| | | M1 | 0.02 | 0.28 | 0.29 | 3.76 | 0.42 | 0.45 | 0.20 | 2.28 |



**Table 4.** Ranks of the inverse tests with various extra constraints evaluated against G2, G3, G4, and G5 observations (OBS). Mass loading (ML): MA, integrating from surface to cloud top; M0, calculated for a single layer where the cloud top height resides; M1, integrating over three layers centered at the cloud top layer. Extra zero observation constraints: H, with clear pixels; T, with clear column above ash cloud; B, with clear column below ash cloud. Ash cells are assimilated in all inverse cases. Satellite data at both G1 and G2 are used for all cases listed here.

| OBS | ML | GDAS | | | | ECMWF | | | |
|---|---|---|---|---|---|---|---|---|---|
| | | - | H | H+T | H+T+B | - | H | H+T | H+T+B |
| G2 | MA | 2.61 | 2.26 | 2.00 | - | 2.81 | 2.50 | 2.20 | - |
| | M0 | 2.58 | 2.03 | 1.65 | 1.17 | 2.87 | 2.46 | 1.82 | 1.22 |
| | M1 | 2.66 | 2.27 | 2.17 | 1.81 | 2.90 | 2.63 | 2.54 | 2.00 |
| G3 | MA | 2.72 | 2.38 | 2.04 | - | 2.80 | 2.53 | 2.17 | - |
| | M0 | 2.77 | 2.21 | 1.74 | 1.37 | 2.85 | 2.61 | 1.83 | 1.33 |
| | M1 | 2.77 | 2.36 | 2.25 | 1.88 | 2.80 | 2.61 | 2.56 | 2.06 |
| G4 | MA | 2.96 | 2.64 | 2.23 | - | 2.80 | 2.50 | 2.37 | - |
| | M0 | 2.96 | 2.45 | 1.83 | 1.40 | 2.90 | 2.81 | 2.03 | 1.38 |
| | M1 | 3.02 | 2.62 | 2.51 | 2.05 | 2.78 | 2.74 | 2.72 | 2.17 |
| G5 | MA | 2.34 | 2.05 | 1.73 | - | 2.33 | 2.28 | 2.01 | - |
| | M0 | 2.31 | 2.08 | 1.52 | 1.05 | 2.17 | 2.32 | 1.77 | 1.05 |
| | M1 | 2.32 | 2.04 | 1.96 | 1.70 | 2.28 | 2.38 | 2.39 | 1.81 |

**Table 5.** Ranks against G2–G5 for HYSPLIT simulations after assimilating various combinations of observation inputs. Model counterparts of the satellite mass loadings are calculated using "M1" option, i.e. integrating over three layers centered at the cloud top layer. Only ash cells are assimilated for all the inverse cases listed here. "()" indicates that the observations have been assimilated.

| Inputs | GDAS | | | | ECMWF | | | |
|---|---|---|---|---|---|---|---|---|
| | G2 | G3 | G4 | G5 | G2 | G3 | G4 | G5 |
| G2 | (2.70) | 2.69 | 2.86 | 2.27 | (2.90) | 2.76 | 2.76 | 2.29 |
| G1,G2 | (2.66) | 2.77 | 3.02 | 2.32 | (2.90) | 2.80 | 2.78 | 2.28 |
| G3 | 2.59 | (3.16) | 2.89 | 2.20 | 2.43 | (3.07) | 2.78 | 2.10 |
| G2,G3 | (2.69) | (2.94) | 2.94 | 2.26 | (2.76) | (2.91) | 2.81 | 2.23 |
| G1,G2,G3 | (2.61) | (2.93) | 2.96 | 2.28 | (2.77) | (2.98) | 2.86 | 2.20 |




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
