# Peer review of "Improving volcanic ash predictions with the HYSPLIT dispersion model by assimilating MODIS satellite retrievals"

_Atmospheric Chemistry and Physics, 2016_

## Referee Comment (RC1) · Anonymous Referee #2 · 7 Nov 2016

Review of 'Improving volcanic ash predictions with the HYSPLIT dispersion model by assimilating MODIS satellite retrievals' by Chai et al

General comments:

The paper presents an inversion method for diagnosing emission rates for volcanic eruptions, applies it to the 2008 Kasatochi eruption, and conducts a range of sensitivity tests to assess various modifications to the approach. This adds value to previous similar studies through testing a variety of plausible approaches and by applying the method to a volcanic eruption that has not been studied in this way before. The latter aspect is especially welcome as previous studies have only used a very small number of eruptions and it is unclear how widely applicable the conclusions are. The paper is

suitable for publication as a discussion in ACP both in terms of scope and in terms of scientific soundness.

Specific comments:

1) It would be nice to have a little more discussion about the meaning and limitations of satellite derived ash cloud top. In many retrieval systems, for optically thin clouds, this may be more like the mean ash cloud height.

2) Page 5, lines 11-13: I guess the significance of the different sizes is that the particles have a fall speed – it would be good to clarify if that is correct. Also, while the size distribution chosen seems very sensible, it would be good to say what the basis of the distribution is, e.g. perhaps it's based on some particular measurements. If it's just expert judgement, that's fine.

3) The convention that cloud top height is infinite when there is no ash cloud (p 6, line 8-9 and p 13, line 31) seems strange. If one thinks of it as the height above which there is no ash, then zero seems more appropriate than infinity. In any case I think the convention is not needed in the paper – would anything change if infinity was replaced by zero? If not it would be simpler to just talk about no ash regions and not mention a cloud top height for such regions.

4) I think that, if the zero ash observed values are not used (i.e. from ash free regions or values above and below the ash cloud), emissions which don't contribute to the chosen model diagnostics because they are much higher than the observed ash top, are not constrained by the observations. These emissions will then be set to the *a priori* values. This only works because the *a priori* is chosen to be small. Assuming this is correct, it would be good to explain this.

5) Assuming a single model layer for the model diagnostic and imposing zero values above and below this layer will clearly give results that are sensitive to errors in the observed ash cloud top. E.g. if the top is in error and the winds at the true and

observed heights are in different directions, the method will not work very well (as is seen). I think it would be useful (but not essential) to give more discussion of these sorts of aspects rather than just presenting the results and noting which methods work best.

6) Page 14, line 27: The idea of a cylindrical source is interesting, but readers won't be able to assess this without a little more information about the Kasatochi eruption. In particular, was there a significant umbrella cloud generated by the eruption? Probably this is discussed by Crawford et al (2016), but a few extra words would help the reader.

Technical corrections:

These are mainly requests for clarification or minor corrections.

7) Some of the options are not easy to understand from the presentation in the abstract (lines 10-17). This may be inevitable to some extent given the space restrictions, but it would be nice to give a little more information. For example I think the 'three options' are not really options for the matching method but for the choice of model diagnostic, so that, in the 'integrating over three layers' option, the model result over three model layers is compared with the satellite total column – there's no attempt to retrieve column load over just three model layers from the satellite. Also when using the three model layers option and enforcing no ash above/below the observed ash top, I assume that this is not enforced in the top/bottom of the three layers, so that 'above/below the cloud' is interpreted in relation to the chosen model cloud diagnostic. These aspects are clearer on page 6, but the last aspect is still not completely unambiguous.

8) Identifying 'no ash detected' with 'ash free' (p 3, lines 22-24 and p 6, line 8) is explained later as being applicable to Kasatochi where there is little meteorological cloud (p 6, lines 21-24), but is not necessarily applicable in general. It's worth considering if something can be said earlier to avoid readers thinking that the authors have made an incorrect identification.

9) Page 7, lines 22-24: It sounds as though these zero values are used in all the inversions, but in fact this is only true in some of the approaches used. Might also be worth clarifying on p 9 whether the zero values are used in fig 3 and 4 (and also in section 4.2). It becomes clear in section 4.3 that the zeros are considered in 4.3 and hence weren't included before, but this could be made clear earlier.

10) Page 7, line 30: I guess the approach used and the alternative described are equivalent, in that e.g. $q_{ij} - q_{ij}^b$ in (1) is replaced by $\exp(l_{ij}) - q_{ij}^b$ with $l = \log q$ and with $l_{ij}$ being adjusted, so the same quantity is being minimised (rather than replacing $q_{ij} - q_{ij}^b$ in (1) by $l_{ij} - \log(q_{ij}^b)$). Could avoid any doubt here by saying that the alternative method is equivalent or should give the same answer or is an alternative way to solve the same mathematical problem.

11) It might be clearer to emphasise a bit more that ash cloud top usually means observed ash cloud top (and not the unknown true ash cloud top or a model value). E.g. fig 2 caption and page 9, line 24.

12) Page 11, line 15: I don't think it is correct to say that the authors find the method which *improved* the forecasts the most, since they don't compare with an approach that doesn't use any inversion modelling. Instead they just find out which method is best.

---

## Referee Comment (RC2) · Anonymous Referee #1 · 10 Nov 2016

General comments

The paper "Improving volcanic ash predictions with the HYSPLIT dispersion model by assimilating MODIS satellite retrievals" by Chai et al. discusses the inversion results for the Kasatochi 2008 event [ash] using a combination of the HYSPLIT model, providing the TCMs, and MODIS satellite data. Although inverse modelling studies for volcanic eruptions following similar methodologies are not new, the results for this eruption and the sensitivity studies presented here provide useful information to the reader and therefore supports its publishing in ACPD.

Specific comments

1) Page 2, first paragraph: although not strictly needed, it would be good that it includes additional references on the impacts on the aviation industry as well as references on the residence times of the fine ash fraction.

2) Page 5, line 13: how was the particle size distribution estimated? It would be good to know the rationale behind the selection of the four bins, their sizes and percentage of distributions. Is that based on measurements? Estimates from another study? Please comment or add reference. This may include a comment on why the largest one considered is 20 um in relation to the satellite sensitivities and what limitations will this pose when it is the whole fine ash fraction (< 63 um) that may potentially affect aviation.

3) Section 2.1 and Section 2.5 line 26: it is clear that the observation uncertainties play a significant role in the inversion. It would be valuable to add more discussion on the uncertainties and errors in the observations in either of the sections (and explain how the estimate of the observational errors are assumed to be 0.5 x am + 0.3 g/m2) with special emphasis on the cloud top since this parameter is used to define the three options for model to observations adjustment. This is obviously of importance for the second option, where the cloud top is critical and fixes the only model level that will be used in the matching.

3) Section 2.3 and 3: the definition of the three options to match the model to observations clearly affects the results. It seems that using the three layers approach, whereby ash above the cloud top is allowed, improves the results. Have the authors considered using option 1 but also allowing that the layer above the cloud top is also considered?

4) Section 2.5, line 20. What is the basis for the selection of this a priori emission rate and vertical distribution?

5) Setion3: before starting to discuss Figure 3 (line14) please add (move) lines 24 to 25 so that the user knows what simulations (using GDAS or ECMWF data) the authors are referring to.

6) Section 3, line 19: why do the authors finally use the a priori error variances of 2.8 x 10**5 kg/s? I see no justification in the text and that would mean that either of the two error variances shown would be usable.

7) Section 4.2 and following: in line 29 the authors state that Stein et al. (2015b) estimated the uncertainties for the Rank to be of 0.1. However, in all the tables and most of the discussion is based on those numbers, we see the ranks (and all the statistical metrics) to have to significant decimals. How can then we judge the performance of the different MA, M0 and M1 options when often is the second decimal that varies?

8) Could the authors give a better justification of why the zero mass loading pixels correspond to infinite cloud top heights?

9) Comparing the simulations with assimilated data (including G2) to G2 observations does not provide real insight since we are comparing assimilated results with the data used in the assimilation procedure. I think it is more useful to base the discussion comparing with G3 onwards if G1 and G2 are assimilated and with G2 onwards if only G1 is assimilated.

Technical corrections

1) Figure 2: please add in the caption that those are the TCMs obtained with the GDAS input data
* * *

---

## Author Comment (AC1) · 19 Jan 2017

**Improving volcanic ash simulations with the HYSPLIT dispersion model by assimilating MODIS satellite retrievals**

Tianfeng Chai, Alice Crawford, Barbara Stunder, Michael J. Pavolonis, Roland Draxler, and Ariel Stein

**Response to Reviewers' comments**

January 19, 2017

**Responses to the comments of referee #1:**

**General comments:**

*The paper "Improving volcanic ash predictions with the HYSPLIT dispersion model by assimilating MODIS satellite retrievals" by Chai et al. discusses the inversion results for the Kasatochi 2008 event [ash] using a combination of the HYSPLIT model, providing the TCMs, and MODIS satellite data. Although inverse modelling studies for volcanic eruptions following similar methodologies are not new, the results for this eruption and the sensitivity studies presented here provide useful information to the reader and therefore supports its publishing in ACPD.*

We thank the referee for thoroughly reading the manuscript and providing valuable comments. Point-by-point responses to the referee's specific comments are given below.

**Specific comments:**

1) *Page 2, first paragraph: although not strictly needed, it would be good that it includes additional references on the impacts on the aviation industry as well as references on the residence times of the fine ash fraction.*

   The following two papers are added as the references on the impacts of volcanic ash on the aviation industry. In addition, three other papers (Wilson et al., 2011; Horwell and Baxter, 2006; Wilson et al., 2012) are included as the references on the other impacts of volcanic ash.

   *Prata, A. J. and Tupper, A.: Aviation hazards from volcanoes: the state of the science, NATURAL HAZARDS, 51, 239–244, doi10.1007/s11069-009-9415-y, 2009.*

   *Gordeev, E. I. and Girina, O. A.: Volcanoes and their hazard to aviation, HERALD OF THE RUSSIAN ACADEMY OF SCIENCES, 84, 1–8,*

Rose and Durant(2009) is included as a reference on the residence times of the fine ash.

*Rose, W. I. and Durant, A. J.: Fine ash content of explosive eruptions, J. Volcanol. Geotherm. Res., 186, 32–39, doi10.1016/j.jvolgeores.2009.01.010, 2009.*

2) *Page 5, line 13: how was the particle size distribution estimated? It would be good to know the rationale behind the selection of the four bins, their sizes and percentage of distributions. Is that based on measurements? Estimates from another study? Please comment or add reference. This may include a comment on why the largest one considered is 20 um in relation to the satellite sensitivities and what limitations will this pose when it is the whole fine ash fraction (< 63 um) that may potentially affect aviation.*

The particle size distribution was originally used in the NOAA ARL VAFTAD model based on aircraft samplings of Mount St. Helens and Redoubt Volcano ash clouds. Several grain size distributions were tested by Webley et al.(2009) and were found to cause little effect in ash cloud simulation. The following has been added to the manuscript.

*The same particle size distribution was originally used in the NOAA ARL VAFTAD model (Heffter and Stunder,1993). Webley et al.(2009) evaluated the sensitivity of the grain size distribution on the modeled ash cloud and found that this pre-defined distribution is sufficient for HYSPLIT volcanic ash simulation. MODIS effective particle radii ($r_{eff}$) are retrieved to describe the ash particle size distributions. However, $r_{eff}$ greater than 15–20$\mu$m are not retrieved since the retrievals cannot be performed reliably when $r_{eff}$ exceeds 15$\mu$m (Pavolonis et al., 2013).*

*Heffter, J. and Stunder, B.: Volcanic ash forecast transport and dispersion (VAFTAD) model, Weather and Forecasting, 8, 533–541, doi:10.1175/1520-0434(1993)008<0533:VAFTAD>2.0.CO;2, 1993.*

*Webley, P. W., Stunder, B. J. B., and Dean, K. G.: Preliminary sensitivity study of eruption source parameters for operational volcanic ash cloud transport and dispersion models - A case study of the August 1992 eruption of the Crater Peak vent, Mount Spurr, Alaska, J. Volcanol. Geotherm. Res., 186, 108–119, doi:10.1016j.jvolgeores.2009.02.012, 2009.*

3) *Section 2.1 and Section 2.5 line 26: it is clear that the observation uncertainties play a significant role in the inversion. It would be valuable to add more discussion on the uncertainties and errors in the observations in either of the sections (and explain how the estimate of the observational errors are assumed to be 0.5 x am + 0.3 g/m2) with special emphasis on the cloud top since this parameter is used to define the three options for model to observations adjustment. This is obviously of importance for the*

*second option, where the cloud top is critical and fixes the only model level that will be used in the matching.*

The following has been added in Section 2.5 to provide explanation for the observational error assumption.

*Dubuisson et al. (2014) studied the remote sensing of volcanic ash plumes from SE-VIRI, MODIS and IASI instruments. The total uncertainty in MODIS mass loading resulted from errors in the input atmospheric parameters such as ash layer altitude, particle size distribution, and particle composition was estimated to be $\sim 50\%$. Their inter-comparison among six satellite configurations shows a standard deviation of 0.3 $g/m^2$ for the mean mass loading estimates. In this study, the observational errors are estimated using $\epsilon_m = 0.50 \times a_m^o + 0.3$ $g/m^2$.*

*Dubuisson, P., Herbin, H., Minvielle, F., Compiegne, M., Thieuleux, F., Parol, F., and Pelon, J.: Remote sensing of volcanic ash plumes from thermal infrared: a case study analysis from SEVIRI, MODIS and IASI instruments, Atmos Meas Tech., 7, 359–371, doi10.5194/amt-7-359-2014, 2014.*

The following discussion on the cloud top uncertainties is added in the first paragraph in Section 2.3, after the second model-to-observation matching option is introduced.

*However, the retrieved cloud top heights are associated with uncertainties. Pavolonis et al. (2013) showed that the retrieved cloud top height had a low bias of 0.77 km relative to lidar. Crawford et al. (2016) compared MODIS cloud top height retrievals with CALIOP vertical profiles of the same event. In general, the MODIS top heights agree well with the top aerosol level indicated by CALIOP profiles but can be off by several kilometers. When CALIOP shows two levels of ash, the MODIS top height falls between them. In addition, the cloud top height retrievals typically lie in the middle of thick ash cloud layers rather than at the top (Pavolonis et al., 2013). To compensate for such uncertainties in ash cloud top height position, the third option is designed to integrate model volcanic ash concentrations over three model layers, i.e. from one layer below to one layer above the cloud top layer.*

*Section 2.3 and 3: the definition of the three options to match the model to observations clearly affects the results. It seems that using the three layers approach, whereby ash above the cloud top is allowed, improves the results. Have the authors considered using option 1 but also allowing that the layer above the cloud top is also considered?*

We tested such an option, among several others (such as integrating over all layers or 5 layers) that are not presented in the manuscript. Based on the tests where only G2 observations were assimilated, the results using this option is not significantly different

from the results using option 1. Although we decided to exclude this option mainly for brevity in presentation, we believe this option is worth further investigation in the future.

4) *Section 2.5, line 20. What is the basis for the selection of this a priori emission rate and vertical distribution?*

The constant value and uniform vertical distribution are for simplicity. The small *a priori* emission rate is chosen to avoid unrealistic release rates at time-locations that the observations do not provide any information to modify them. Explanation has been added to text, as shown below.

*For emission points at which the release generates no simulated ash corresponding to any of the assimilated observations, the first guesses remain unchanged. To avoid unrealistic release rates for such emission points, we chose a small constant emission rate of $10^4$ g/hr ($\approx 2.8 \times 10^{-3} kg/s$) at all hours and layers as the first guess.*

5) *Section3: before starting to discuss Figure 3 (line14) please add (move) lines 24 to 25 so that the user knows what simulations (using GDAS or ECMWF data) the authors are referring to.*

This statement has been moved as suggested.

6) *Section 3, line 19: why do the authors finally use the a priori error variances of 2.8 x 10\*\*5 kg/s? I see no justification in the text and that would mean that either of the two error variances shown would be usable.*

The statement has been extended (shown below) to include a brief justification for the *a priori* error variances.

*Note that a larger a priori term with smaller a priori error variances in Equation 1 typically helps the minimization procedure in the emission inversion. Since the results using the two a priori errors are similar, the a priori error variances are set as $\sigma_{ij} \approx 10^{12}$ g/hr ($\approx 2.8 \times 10^5$ kg/s) in the following tests.*

7) *Section 4.2 and following: in line 29 the authors state that Stein et al. (2015b) estimated the uncertainties for the Rank to be of 0.1. However, in all the tables and most of the discussion is based on those numbers, we see the ranks (and all the statistical metrics) to have to significant decimals. How can then we judge the performance of the different MA, M0 and M1 options when often is the second decimal that varies?*

Stein et al. (2015b) estimated the uncertainties of the Rank as 0.08, 0.08, 0.09, 0.08, 0.11, and 0.07 for 6 different tracer releases. The uncertainties of the Rank for the current application could vary but they are not expected to be too different. Thus, two significant decimals are presented and a difference of smaller than 0.1 in Rank may still be significant. While we agree that the performance differences with MA, M0 and M1 options are mostly small, we carefully stated that the M1 option is "slightly better" than the other two options in both Abstract and Summary.

The statement on the uncertainties for the Rank has been clarified (shown below).

*Using HYSPLIT ensembles, Stein et al. (2015b) estimated the uncertainties of the Rank as 0.08, 0.08, 0.09, 0.08, 0.11, and 0.07 for 6 different tracer releases. The uncertainties of the Rank for the current application could vary but they are not expected to be too different.*

8) *Could the authors give a better justification of why the zero mass loading pixels correspond to infinite cloud top heights?*

"Infinite cloud top heights" were used to indicate that the modeled mass loadings integrated from surface to the highest level possible should yield zero mass loadings. As it can be ambiguous, the statement in Section 2.3 (2nd sentence of the 2nd paragraph), "This is equivalent to zero mass loading and infinite cloud top height", is changed to "This is equivalent to zero mass loading for the entire atmospheric column at such a location."

In addition, the first sentence in Section 4.3, "ash-free regions indicate zero mass loadings and infinite cloud top heights", is changed to "ash-free regions indicate zero mass loadings for the entire atmospheric columns."

9) *Comparing the simulations with assimilated data (including G2) to G2 observations does not provide real insight since we are comparing assimilated results with the data used in the assimilation procedure. I think it is more useful to base the discussion comparing with G3 onwards if G1 and G2 are assimilated and with G2 onwards if only G1 is assimilated.*

We agree that comparing model results with un-assimilated data will be more useful. In fact, most of the discussion is based on such comparison. For the same reason, no comparison with G1 is listed or discussed. As G2 is not assimilated in some cases, including the comparison with G2 still provides some insight. For instance, in Section 4.2 we found, "If only G2 observations were assimilated, the model performance would be expected to peak

when compared against G2. However, as both G1 and G2 observations are assimilated, this is no longer true."

**Technical corrections**

1) *Figure 2: please add in the caption that those are the TCMs obtained with the GDAS input data*

It has been added. Now the caption reads "Averaged TCMs using GDAS meteorological data with three different options ...".

---

## Author Comment (AC2) · 19 Jan 2017

**Responses to the comments of referee #2:**

*The paper presents an inversion method for diagnosing emission rates for volcanic eruptions, applies it to the 2008 Kasatochi eruption, and conducts a range of sensitivity tests to assess various modifications to the approach. This adds value to previous similar studies through testing a variety of plausible approaches and by applying the method to a volcanic eruption that has not been studied in this way before. The latter aspect is especially welcome as previous studies have only used a very small number of eruptions and it is unclear how widely applicable the conclusions are. The paper is suitable for publication as a discussion in ACP both in terms of scope and in terms of scientific soundness.* We thank the reviewer for reading the manuscript thoroughly and appreciate the insightful comments and constructive suggestions. The specific comments have been addressed below.

**Specific comments:**

1) *It would be nice to have a little more discussion about the meaning and limitations of satellite derived ash cloud top. In many retrieval systems, for optically thin clouds, this may be more like the mean ash cloud height.*

   The following discussion on the cloud top uncertainties is added in Section 2.3.

   *However, the retrieved cloud top heights are associated with uncertainties. Pavolonis et al. (2013) showed that the retrieved cloud top height had a low bias of 0.77 km relative to lidar. Crawford et al. (2016) compared MODIS cloud top height retrievals with CALIOP vertical profiles of the same event. In general, the MODIS top heights agree well with the top aerosol level indicated by CALIOP profiles but can be off by several kilometers. When CALIOP shows two levels of ash, the MODIS top height falls between them. In addition, the cloud top height retrievals typically lie in the middle of thick ash cloud layers rather than at the top (Pavolonis et al., 2013).*

2) *Page 5, lines 11-13: I guess the significance of the different sizes is that the particles have a fall speed – it would be good to clarify if that is correct. Also, while the size distribution chosen seems very sensible, it would be good to say what the basis of the distribution is, e.g. perhaps it's based on some particular measurements. If it's just expert judgement, that's fine.*

   This has been clarified by adding "with different fall speeds according to Stokes's law (Heffter and Stunder, 1993)" after the sentence "In each simulation, particles of four different sizes are released as different pollutants".

The basis of the chosen distribution is also provided with the following text added to the manuscript.

*The same particle size distribution was originally used in the NOAA ARL VAFTAD model (Heffter and Stunder,1993). Webley et al.(2009) evaluated the sensitivity of the grain size distribution on the modeled ash cloud and found that this pre-defined distribution is sufficient for HYSPLIT volcanic ash simulation. MODIS effective particle radii ($r_{eff}$) are retrieved to describe the ash particle size distributions. However, $r_{eff}$ greater than 15–20µm are not retrieved since the retrievals cannot be performed reliably when $r_{eff}$ exceeds 15µm (Pavolonis et al., 2013).*

*Heffter, J. and Stunder, B.: Volcanic ash forecast transport and dispersion (VAFTAD) model, Weather and Forecasting, 8, 533–541, doi:10.1175/1520-0434(1993)008<0533:VAFTAD>2.0.CO;2, 1993.*

*Webley, P. W., Stunder, B. J. B., and Dean, K. G.: Preliminary sensitivity study of eruption source parameters for operational volcanic ash cloud transport and dispersion models - A case study of the August 1992 eruption of the Crater Peak vent, Mount Spurr, Alaska, J. Volcanol. Geotherm. Res., 186, 108–119, doi:10.1016j.jvolgeores.2009.02.012, 2009.*

3) *The convention that cloud top height is infinite when there is no ash cloud (p 6, line 8-9 and p 13, line 31) seems strange. If one thinks of it as the height above which there is no ash, then zero seems more appropriate than infinity. In any case I think the convention is not needed in the paper – would anything change if infinity was replaced by zero? If not it would be simpler to just talk about no ash regions and not mention a cloud top height for such regions.*

"Infinite cloud top heights" were used to indicate that the modeled mass loadings integrated from surface to the highest level possible should yield zero mass loadings. We agree that it is better to just talk about no ash regions without mentioning the cloud top height. This sentence has been changed to "This is equivalent to zero mass loading for the entire atmospheric column at such a location."

In addition, the first sentence in Section 4.3, "ash-free regions indicate zero mass loadings and infinite cloud top heights", is changed to "ash-free regions indicate zero mass loadings for the entire atmospheric columns."

4) *I think that, if the zero ash observed values are not used (i.e. from ash free regions or values above and below the ash cloud), emissions which don't contribute to the chosen model diagnostics because they are much higher than the observed ash top, are not constrained by the observations. These emissions will then be set to the a priori values.*

8

*This only works because the a priori is chosen to be small. Assuming this is correct, it would be good to explain this.*

It is correct. The small *a priori* emission rate is chosen to avoid unrealistic release rates at time-locations that the observations do not provide any information to modify them. The following explanation has been added in Section 2.5 for the chosen *a priori* emission rate.

*For emission points at which the release generates no simulated ash corresponding to any of the assimilated observations, the first guesses remain unchanged. To avoid unrealistic release rates for such emission points, we chose a small constant emission rate of $10^4$ g/hr ($\approx 2.8 \times 10^{-3} kg/s$) at all hours and layers as the first guess.*

5) *Assuming a single model layer for the model diagnostic and imposing zero values above and below this layer will clearly give results that are sensitive to errors in the observed ash cloud top. E.g. if the top is in error and the winds at the true and observed heights are in different directions, the method will not work very well (as is seen). I think it would be useful (but not essential) to give more discussion of these sorts of aspects rather than just presenting the results and noting which methods work best.*

More discussion has been added. Now the end of Section 4.3 reads,

*... and 2.28 to 2.38, respectively. Enforcing the extra constraints of the ash-free regions makes the inversion results very sensitive to the transport errors since the HYSPLIT simulated ash plume outside the MODIS ash cloud starts to affect the emission inversion results. Table 4 shows that the emission inversion with extra constraints of clear pixels using ECMWF data performs better than using GDAS data except a single case with the MA option against G4.*

*Adding the extra constraints of a clear column above the ash cloud again generally causes a decrease in Rank. An exception is the ECMWF case with the M1 option (three model layers used for mass loading calculation) in which the extra "top" constraint results in a marginally better predictions evaluated against G5 (Rank 2.39 versus 2.38). It is found that the ECMWF cases perform better than all their GDAS counterparts after adding the "top" constraints. When the constraints of clear column below ash cloud are further added for the M0 and M1 options, the ranks decrease significantly, especially for the M0 option in which a single model layer is used to construct the model mass loadings. Clearly, model and observation uncertainties have to be carefully addressed to take advantage of the extra constraints in order to benefit the emission inversion. This requires further investigation in future studies.*

9

6) *Page 14, line 27: The idea of a cylindrical source is interesting, but readers won't be able to assess this without a little more information about the Kasatochi eruption. In particular, was there a significant umbrella cloud generated by the eruption? Probably this is discussed by Crawford et al (2016), but a few extra words would help the reader.*

The following sentence is added.

*Waythomas et al.(2010) showed that the source area was quite broad with a width about 75 km from 06Z to 10Z on August 8, 2008.*

**Technical corrections:**

These are mainly requests for clarification or minor corrections.

7) *Some of the options are not easy to understand from the presentation in the abstract (lines 10-17). This may be inevitable to some extent given the space restrictions, but it would be nice to give a little more information. For example I think the 'three options' are not really options for the matching method but for the choice of model diagnostic, so that, in the 'integrating over three layers' option, the model result over three model layers is compared with the satellite total column – there's no attempt to retrieve column load over just three model layers from the satellite. Also when using the three model layers option and enforcing no ash above/below the observed ash top, I assume that this is not enforced in the top/bottom of the three layers, so that 'above/below the cloud' is interpreted in relation to the chosen model cloud diagnostic. These aspects are clearer on page 6, but the last aspect is still not completely unambiguous.*

Thanks for the suggestion to use "model diagnostic". We modified text at a couple of places.

In Abstract, "Because the satellite retrievals include the ash cloud top height but not the bottom height, three options for matching the model concentrations to the observed mass loadings are tested" is changed to "Because the satellite retrievals include the ash cloud top height but not the bottom height, there are different model diagnostic choices when comparing the model results with the observed mass loadings. Three options are presented and tested."

The title of Section 2.3 is changed to "Model diagnostic" from "Matching model to observations".

To clarify what 'above/below the cloud' means, the follow text is added in Section 4.3.

*Note that the term "above or below ash cloud" is in relation to the chosen model cloud diagnostic. For instance, if M1 option is chosen, above and below ash cloud constraints*

10

*are enforced over the model layers outside the three ash layers.*

8) *Identifying 'no ash detected' with 'ash free' (p 3, lines 22-24 and p 6, line 8) is explained later as being applicable to Kasatochi where there is little meteorological cloud (p 6, lines 21-24), but is not necessarily applicable in general. It's worth considering if something can be said earlier to avoid readers thinking that the authors have made an incorrect identification.*

The following text has been added to Section 2.1.

*Note that the ash-free regions do not apply to regions with missing ash mass loadings due to meteorological cloud or other reasons.*

9) *Page 7, lines 22-24: It sounds as though these zero values are used in all the inversions, but in fact this is only true in some of the approaches used. Might also be worth clarifying on p 9 whether the zero values are used in fig 3 and 4 (and also in section 4.2). It becomes clear in section 4.3 that the zeros are considered in 4.3 and hence weren't included before, but this could be made clear earlier*

The sentence (Page 7, lines 22-24) is changed to "The observations here refer to both the volcanic ash mass loadings for the ash cloud and the zero values for the ash-free regions which are later included as extra constraints in Section 4.3."

Clarifications are made at all the three places the referee suggested.

10) *Page 7, line 30: I guess the approach used and the alternative described are equivalent, in that e.g. $q_{ij} - q_{ij}^b$ in (1) is replaced by $exp(l_{ij}) - q_{ij}^b$ with $l = log\,q$ and with $l_{ij}$ being adjusted, so the same quantity is being minimised (rather than replacing $q_{ij} - q_{ij}^b$ in (1) by $l_{ij} - log(q_{ij}^b)$. Could avoid any doubt here by saying that the alternative method is equivalent or should give the same answer or is an alternative way to solve the same mathematical problem.*

Yes, it is an alternative way to solve the problem. The following statement has been added.

*As they solve the same mathematical problem, these two ways are expected to arrive at the same answer with enough iterations.*

11) *It might be clearer to emphasise a bit more that ash cloud top usually means observed ash cloud top (and not the unknown true ash cloud top or a model value). E.g. fig 2 caption and page 9, line 24.*

Changes have been made at several places (E.g., Abstract, Fig 2 caption) to emphasize that ash cloud top usually means observed ash cloud top. Note that the statement (page 9, line 24) the referee mentioned is also modified. But it has been moved up to the 2nd paragraph in Section 3, as suggested by the other referee.

12) *Page 11, line 15: I don't think it is correct to say that the authors find the method which improved the forecasts the most, since they don't compare with an approach that doesn't use any inversion modelling. Instead they just find out which method is best.*

We removed the later part of the sentence. Now it reads "A series of tests were performed to find the best inverse modeling setup."

---

## Author Comment (AC3) · 19 Jan 2017

The manuscript has been revised. Please see the supplement pdf file.

Please also note the supplement to this comment:
http://www.atmos-chem-phys-discuss.net/acp-2016-750/acp-2016-750-AC3-supplement.pdf